# SOUP TO GO: MITIGATING FORGETTING DURING CONTINUAL LEARNING WITH MODEL AVERAGING

## ABSTRACT

In continual learning with pretrained large language models (LLMs), where data from instruction fine-tuning (IFT) tasks arrives in a sequence, fine-tuning on later tasks will often lead to performance degradation on earlier tasks. This is especially pronounced when the IFT tasks come from diverse domains. In this setting, how can we mitigate catastrophic forgetting of earlier tasks and retain what the LLM has learned? Inspired by a classical continual learning method—L2 penalty to previous weights—we propose Sequential Fine-tuning with Averaging (SFA), a method that merges models with earlier checkpoints trained on previous tasks during the course of training. SOTA approaches typically maintain a data buffer of past tasks or impose a penalty at each gradient step. In contrast, our method achieves comparable results *without* the need to store past data, or multiple copies of parameters for each gradient step. Furthermore, our method outperforms penalty methods like L2 and Elastic Weight Consolidation, as well as other common merging techniques such as Task Arithmetic, and TIES Merging. Finally, we show that using our method, a single model can simultaneously perform well on a range of fine-tuning tasks in diverse domains, including Math, Law and Code.

## 1 INTRODUCTION

Fine-tuning large language models (LLMs) on new tasks often leads to catastrophic forgetting: the rapid degradation of performance on previously learned tasks (Scialom et al., 2022; Lesort et al., 2019; Delange et al., 2021; Belouadah et al., 2021; Luo et al., 2023). This poses a major challenge for continual learning scenarios, where data comes in a stream of sequences of tasks that may not reappear. As such, we are in need of fine-tuning procedures that would allow LLMs to continually adapt to new knowledge without sacrificing past abilities.

Previous work has analyzed catastrophic forgetting of different types of information, as well as the impact of scale. Scialom et al. (2022) explain that LLMs can perform worse on past fine-tuning tasks as they learn new ones. Furthermore, Luo et al. (2023) show a model can also forget general knowledge, not specific to a single past task. Finally, forgetting also grows in severity as model size increases (Luo et al., 2023). Existing state-of-the-art approaches to mitigate forgetting primarily focus on modifying the training data used in fine-tuning. These methods either maintain a data buffer of past tasks (Robins, 1995; Lopez-Paz & Ranzato, 2022; de Masson d'Autume et al., 2019), or generate approximations of past task data for joint training with current tasks (Shin et al., 2017; Mocanu et al., 2016). However, both strategies introduce additional costs. Data buffers increase memory overhead and require careful management, while generating data approximations necessitates extra training and computational resources. Likewise, more classical methods of continual learning that incorporate a penalty directly into training to constrain weights ((Kirkpatrick et al., 2017), L2 penalty) are memory-intensive as they require storing multiple copies of model parameters to be used at each gradient step.

Despite ongoing research into combating forgetting, several key questions remain. What impact does the domain of the fine-tuning tasks have? Specifically, does catastrophic forgetting get even worse when there is a domain shift and if so, by how much? Finally, can we make model-based interventions that can alleviate the cost of storing past data or model parameters, generating new data or doing additional expensive training?

In this paper, we systematically investigate forgetting in pretrained LLMs as they fine-tune on tasks from distinct domains in a continual learning setting. More specifically, we focus on settings where the model sees a sequence of fine-tuning tasks from Law, Math, and Code. In this setting, we analyze the ability of a model pretrained on general language generation to fine-tune to this sequence of new tasks, and track the degradation of the models' existing knowledge on old tasks. By doing so we offer empirical evidence about the nature and rate of forgetting in LLMs during continual learning.

To combat forgetting, we propose Sequential Fine-tuning Averaging (SFA), a novel method that merges the model being trained on a new task with a checkpoint from a previous task during training. This averaged model is then further trained on the new task. By reusing previous checkpoints, SFA promotes knowledge retention across tasks and domains. Our experiments focus on the continual learning settings where data from a sequence of tasks stream in, and only the current task data is available. As such, our solution offers the advantage of not needing to store past data, and simply relies on updating model parameters during fine-tuning with combinations of past and current weights. Furthermore, our solution also does not require training an additional past data generator, because it uses previous model checkpoints as proxies for such data. Our work offers a way forward to obtaining a single model that can perform well on a variety of fine-tuning domains, shedding light on the generalization ability of continual learning LLMs. Our work can be summarized by the following contributions:

- We introduce Sequential Fine-tuning Averaging (SFA), a method for mitigating forgetting by averaging model checkpoints from past tasks during fine-tuning on a new task. This enables the model to retain knowledge on past tasks/domains while learning a new task/domain.
- We show how SFA can be understood as an approximation of a classical continual learning algorithm: applying an L2 penalty between the current model and checkpoints from past tasks. This analysis bridges classical continual learning algorithms such as L2 penalty with commonly used model merging techniques, thus providing intuition for why model merging can be so effective at mitigating forgetting.
- We compare SFA to other techniques for mitigating forgetting across a range of tasks, domains, and models. We show consistent results that across models and tasks/domains, our method achieves comparable results to using a data buffer, while outperforming other model merging techniques, as well as more classical continual learning methods.

## 2 RELATED WORK

**Forgetting and Continual Learning** A large and growing body of literature investigates different aspects of catastrophic forgetting in continual and sequential learning. When the training data consists of disjoint tasks, training classifiers can cause catastrophic forgetting (Rebuffi et al., 2017). Furthermore, if forgetting occurs, it can be tracked during training and is dependent on when examples are seen by the model: models are less likely to remember earlier training examples (Jagielski et al., 2022; Tirumala et al., 2022). Interestingly, forgetting can also occur for general knowledge rather than for specific tasks, and is more severe for larger models (Luo et al., 2023). Lesort et al. (2022) show that overlap between tasks and task repetition in continual learning settings can mitigate catastrophic forgetting of such examples resulting in solutions to forgetting that involve maintaining a data buffer with past data. Such solutions can also be extrapolated to LLMs where continual learning with data repetition can prevent catastrophic forgetting (Scialom et al., 2022). Mitigating forgetting in continual learning can also occur by introducing a penalty in the loss objective. L2 penalty in continual learning constrains the weights of a model as it is learning a new task by introducing a penalty based on the difference between the current and initial model's weights. Similarly, Elastic Weight Consolidation (EWC) (Kirkpatrick et al., 2017) also introduces a penalty to constrain the weights of a model and mitigate increased loss on learned tasks while incorporating the importance of specific weights on learned tasks.

**Model Merging** There exist many techniques and applications for merging multiple models to create a single model with improved generalization on a given set of tasks. Model souping (Wortsman et al., 2022a) involves averaging the parameters of existing models to create a new model. This technique can be applied after training multiple variations of a model on data during a hyperpa-

rameter sweep to combine the models and achieve higher performance than any individual model. Task Arithmetic (Ilharco et al., 2023) involves finding and adding task vectors to create a multi-task model. Wise-FT (Wortsman et al., 2022b) merges the weights of an initial and a fine-tuned model. Our method builds upon these 3 works, but with key differences as described in Section 3.

Additional influential model merging techniques include: Ramé et al. (2023) use a model souping approach to obtain a network with improved out-of-distribution performance by averaging the weights of models fine-tuned on different tasks. TIES (Yadav et al., 2023) only merges influential parameters whose signs are in the direction of greatest movement across the models. Fisher merging (Matena & Raffel, 2022; Dhawan et al., 2023; Jhunjhunwala et al., 2023) requires keeping data from all previous tasks and computing gradients.

Finally for merging different textual domains, Branch-Train-Merge (BTM) (Li et al., 2022) maintains a set of distinct domain models that can be merged and then trained to create new experts.

## 3 METHODOLOGY: SEQUENTIAL FINE-TUNING AVERAGING (SFA)

Our method, Sequential Fine-tuning Averaging (SFA), leverages existing techniques in model merging (Ilharco et al., 2023; Wortsman et al., 2022a;b) to mitigate forgetting in the continual learning setting. In this method, we consider a pretrained LLM that is trained on a sequence of instruction fine-tuning tasks from different domains. While the model is being fine-tuned on the current task, we periodically average the parameters of current model with an earlier checkpoint that resulted from fine-tuning on a previous task. We then continue fine-tuning this new averaged model on the current task.

More precisely, let $\theta_o$ denote the parameters of the network optimized for the last task. Let $\theta_t$ be the parameters of the current model at time $t \leq T$ during fine-tuning on a new task. Then, every $pT$ iterations, we reset the parameters to be a weighted combination of $\theta_o$ and $\theta_t$, where the weighing is determined by a hyperparameter $0 \leq \beta \leq 1$ (default: 0.5).

---

**Algorithm 1** Sequential Fine-tuning Averaging

**Input:** $\theta_o, p, \beta$
Update model parameters $\theta_t$ at each time step $t$
**if** $t \mod pT = 0$ or $t = T$ **then**
  $\theta_t = (\beta)\theta_o + (1 - \beta)\theta_t$

---

By averaging with an optimized model of the last learned task, our method prevents the current model parameters $\theta_t$ from moving significantly from the original model's and thus losing optimal performance on the past task (Section 6.3). In this way, our technique combines the intuition of continual learning with Rehearsal (Robins, 1995), Task Arithmetic (Ilharco et al., 2023) and Wise-FT (Wortsman et al., 2022b). However, unlike Rehearsal-based methods that store data in a buffer, we use a model fine-tuned on a past task/domain. Furthermore, unlike Task Arithmetic, our method merges a past checkpoint of a given model with the current model, rather than the task vectors from individual models. Finally, while our method focuses on merging during actual fine-tuning and across tasks/domains, Wise-FT merges a pretrained and a fine-tuned model. In this way, our work generalizes Wise-FT throughout continual learning. As the number of tasks increases, we continue to average the most recent $\theta_o$, which has high performance on all previous tasks, with the current model parameters $\theta_t$. For example, we merge a model already trained on 2 tasks with the current model training on a third task. We then update $\theta_o$ to be the merged model of all 3 tasks. We find that we are able to preserve performance on all past tasks through continuous averaging.

## 4 CONNECTING CONTINUAL LEARNING METHODS WITH MODEL MERGING

### 4.1 GRADIENT UPDATE COMPARISON

There exist many methods of continual learning that aim to mitigate forgetting of past tasks by constraining training weights using a penalty. This penalty is often used to prevent weights from straying from model weights that perform well on past tasks. Some methods include L1 and L2 penalty, as well as EWC (Kirkpatrick et al., 2017). Typically, these methods add a penalty to an

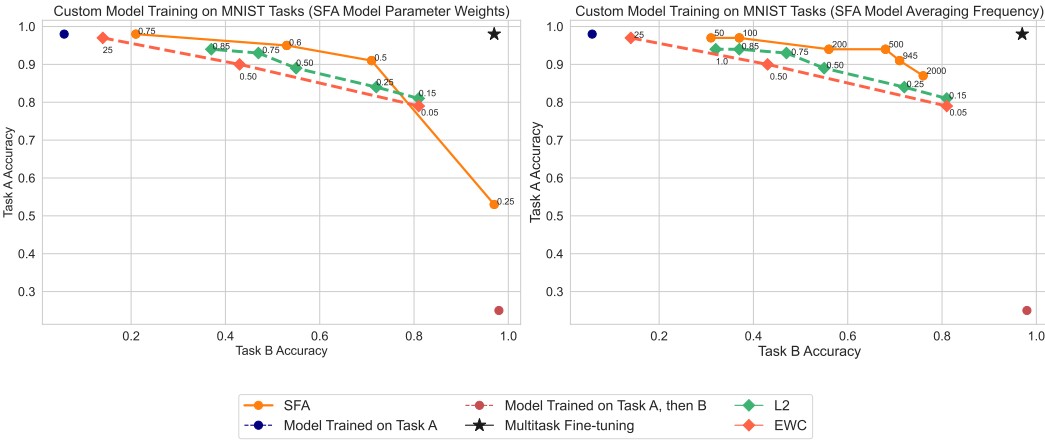

Figure 1: SFA compared against other continual learning methods, where the two tasks (Task A and B) were created by splitting MNIST by label. The accuracy after single-task training, sequential training, and multitask training is also shown. The lines for EWC and L2 are created by varying the coefficient corresponding to each method (and are the same for the left and right plots). **(Left)** visualizes SFA performance under varying $\beta$ coefficient, which determines how much weight is being placed on the initial model. **(Right)** visualizes SFA with varying averaging frequency

existing loss objective for every gradient step. This becomes computationally expensive as models scale for modern day applications, because for each gradient step, multiple copies of model weights have to be loaded in memory to calculate the penalty (e.g. the initial and currently training model), in addition to potential gradients. As such, our work aims to approximate existing continual learning methods with model merging, in order to make them feasible to implement. Specifically, we focus on simplifying and approximating L2 penalty. Consider, starting with $\theta_o$, the model trained on the previous task and $\theta_t$, the model currently being trained on the new task. Calculating the loss with an L2 penalty takes the following form

$$L(\theta) = L_{\text{task}}(\theta_t) + \frac{\lambda}{2}||\theta_t - \theta_o||^2 \tag{1}$$

Updating the model once using the gradient of this loss results in the following:

$$\theta_{t+1} = \theta_t - \eta(\nabla_{\theta_t}L_{\text{task}} + \lambda(\theta_t - \theta_o)) \tag{2}$$

This can be rewritten as:

$$\theta_{t+1} = (1 - \eta\lambda)\theta_t + (\eta\lambda)\theta_o - \eta\nabla_{\theta_t}L_{\text{task}} \tag{3}$$

Now we can compare this to SFA with averaging occurring after each gradient step where the first step updates parameters using only task loss, while the second step updates parameters by averaging the current and initial model:

$$\theta^*_{t+1} = \theta_t - \alpha\nabla_{\theta_t}L_{\text{task}} \tag{4}$$

$$\theta_{t+1} = (1 - \beta)\theta^*_{t+1} + \beta(\theta_o) \tag{5}$$

We can combine these 2 steps to get the following form:

$$\theta_{t+1} = (1 - \beta)(\theta_t - \alpha\nabla_{\theta_t}L_{\text{task}}) + \beta(\theta_o) \tag{6}$$

This is equivalent to:

$$\theta_{t+1} = (1 - \beta)\theta_t + (\beta)\theta_o - \alpha\nabla_{\theta_t}L_{\text{task}}(1 - \beta) \tag{7}$$

As such, Equations 3 and 7 can even be equivalent if $\beta = \eta\lambda$ and $\alpha = \frac{\eta}{(1-\eta\lambda)}$. While in practice, SFA is averaged infrequently, rather than after every gradient step to offer a computational advantage, this implies that it typically is not equivalent to L2-regression. However, the resemblance between Equations 3 and 7, allows SFA to be understood as approximating L2-regression. Similarly, the EWC penalty can also be approximated as a model merging technique (Appendix A.2).

We also show that SFA may have Bayesian motivation because of its similarity to L2-regression (Appendix A.1) We emphasize these connections to bridge commonly used model merging algorithms with classical continual learning ones.

In order to show how our method, SFA, compares with existing continual learning methods, including the one it's approximating, L2 penalty, and EWC, we provide an empirical analysis. In Fig. 1, we train a small, custom neural network on 2 sequential MNIST tasks (Task A and Task B) separated by label introduced in Moriarity (2020). Task A involves labelling the first 5 even numbers, whereas Task B labels the first 5 odd numbers. The blue dot refers to the model after training on Task A, whereas the red dot is additionally trained on Task B without intervention. As such, performance rapidly drops on Task A as the model optimizes for Task B. The solid orange curve refers to SFA where, in Fig. 1 (left) we vary the averaging weight $\beta$ from Algorithm 1 and in Fig. 1 (right) we vary the frequency of averaging in number of batches. As such, placing a higher $\beta$ or lower number of batches before averaging results in a model that performs better on Task A, and vice versa. The green dotted line shows L2 penalty where $\lambda$ (weight on L2 penalty) varies, with a higher $\lambda$ performing better on Task A (and vice versa). Finally, an orange dotted line shows EWC with varying $\lambda$ (weight on EWC penalty) with a higher weight performing better on Task A (and vice versa). L2 penalty outperforms EWC with a better trade off between performance on Task A and B. Interestingly, SFA outperforms both L2 penalty and EWC when hyperparameters are optimized. As such, not only is SFA computationally much cheaper due to infrequent averaging steps, but it is also able to outperform imposing a penalty at every step. Given these optimistic results, we next scale our models and datasets to more realistic fine-tuning scenarios, and apply SFA to directly compare with using a data buffer in continual learning, as well as other model merging methods.

## 5 DATA: CROSS-DOMAIN TASKS

In order to measure and mitigate forgetting, we fine-tune our models on tasks in 3 distinct domains: Law, Math and Code. For each domain, we fine-tune our model on a dataset featuring domain-specific knowledge, as well as unique instruction tasks. For Law, we combine CaseHOLD (Zheng et al., 2021), Terms of Service (ToS) (Lippi et al., 2019; tos, 2023), and Overruling (Zheng et al., 2021) to create a more general Law dataset. For Math, we use MetaMathQA (Yu et al., 2023), and for Code we use MagiCoder110k (Wei et al., 2023). We believe that required task knowledge across these 3 domains is distinct with minimal overlap. As such, we purposefully aim to test our models' ability to generalize across a wide range of knowledge to measure the validity of our method under maximal domain shifts.

**Evaluation Metrics for Data:** In our work, we reference the *forgetting* of various tasks. We define forgetting specific knowledge as a decrease in performance on a given task during evaluation for a model already fine-tuned on the task. For example, if evaluation performance on Task A drops when a model fine-tunes on Task B, given that the model has already fine-tuned on task A, we consider the model to forget Task A. To evaluate performance on our fine-tuning data, we use the metrics and holdout sets described in Table 7.

## 6 RESULTS

### 6.1 MITIGATING FORGETTING FROM CROSS DOMAIN SEQUENTIAL FINE-TUNING USING DATA REHEARSAL

We first confirm that catastrophic forgetting occurs in the scenarios we apply SFA and other baselines to: successive fine-tuning of a pretrained model on instruction tasks (Appendix A.3). In the following sections we focus our analysis of forgetting and its mitigation on pairs of successive instruction fine-tuning tasks with large domain shifts, such as from Math to Code or Math to Law, using datasets outlined in Section 5. This choice of tasks allows us to measure performance with accuracy on downstream tasks instead of with validation loss. By restricting ourselves to pairs of successive tasks, we can clearly quantify the trade off between learning the second task and forgetting the first one by visualizing the results on a plane that measures the accuracy of the first task on the y-axis and the accuracy of the second task on the x-axis. We present our results for sequentially

learning Math and Law with Llama-2 (7B) in Fig. 2 and Math and Law, as well as Math and Code with Pythia (2.8B) in Fig. 3 (see Appendix A.4 for model descriptions).

We first fine-tune our model (Llama 2 (7B) in Fig. 2 and Pythia (2.8B) in Fig. 3) on MetaMathQA to obtain the inital model (dark blue circle). Note the base model performance on the first (second) task is represented by dark green horizontal for Llama 2 (7B), and blue for Pythia (2.8B) (vertical) dashed lines. This initial model improves upon the base on our Math benchmark and is thus higher on the y-axis (performance on first task) while not being significantly different or being worse on the x-axis (performance on the second task which it has not been trained on yet). We then fine-tune the initial model on the second task to obtain the sequential fine-tuning model (red circle). In Fig. 2 the second task is Law while in Fig. 3 the second task is either Law or Code. The sequential fine-tuning model performs really well on the second task (higher on the x-axis) while forgetting almost everything it has learned about the first task (base model level on the y-axis). This movement down and to the right of the initial model (dark blue circle) to the sequential fine-tuning model (red circle) on the task 1 - task 2 performance plane in both Figs. 2 and 3 is emblematic of catastrophic forgetting of an earlier task as the model learns a new task. For reference, the performance of just fine-tuning the base model on the second task is represented by the vertical purple for Law, or green for Code dashed line.

For our upper baseline, we show the results of simultaneously fine-tuning the base model on a mixture of both tasks to obtain the multitask fine-tuning model (black star). This model sits at the upper right of the plane as it does not exhibit forgetting and performs well on both tasks. However, in our continual learning setting where data streams in as a sequence of tasks, this is infeasible.

Rehearsal is a common technique for mitigating forgetting in continual learning. It involves maintaining a buffer of past task data and interleaving it with new task data during fine-tuning (Robins, 1995). We demonstrate the effectiveness of rehearsal in our continual learning setting by further training our initial model (dark blue circle, fine-tuned on Math) on a mixture of 90% task 2 data and 10% of Math data sampled randomly from the full Math dataset. The resulting continual learning (CL) with data buffer model (pink diamond in Figs. 2 and 3) effectively improves on the initial model on task 2 (higher Law performance, i.e. x-axis) while mitigating forgetting (maintains high Math performance i.e. y-axis). Note, this does not work as well for Pythia (2.8B) on Math to Code (Fig. 3, right), we hypothesize that this is because of suboptimal hyperparameters.

A data buffer however, has significant drawbacks: it requires storing data from all previous tasks, leading to rapidly increasing storage costs as the number of tasks grows. It also adds to the training cost, because we must continue to train on tokens from past tasks. Furthermore, maintaining a subset of past data can also threaten data privacy and security (Li et al., 2024). This makes model based mitigations of forgetting appealing.

## 6.2 SFA on Cross Domain Data

Recall that in SFA, we take a model that has already been fine-tuned on Task A, and while fine-tuning on Task B, every $pT$ steps we average the weights with the final model after fine-tuning on Task A and continue fine-tuning on Task B. We evaluate SFA with varying averaging frequency $p$ during cross-domain sequential fine-tuning. Figs. 2 and 3 show that as $p$ decreases, signifying more frequent averaging with the initial model, we observe stronger retention of past domain knowledge (orange curve). By adjusting the averaging frequency ($p$), we control the balance between past and new knowledge retention. This is evident, because as $p$ decreases, the performance on Math (y-axis) increases, indicating stronger retention of task 1. Furthermore, there is minimal loss to the potential learning of task 2 (Law or Code on the x-axis). Notably, when fine-tuning on Math followed by Law, a $p$ of 0.25 yields results comparable to rehearsal (pink diamond), demonstrating that SFA can mitigate forgetting without the need for data buffers. Crucially, our method is able achieve such performance without requiring a data buffer, but just two model checkpoints: the initial one and the current checkpoint throughout fine-tuning.

Additionally, in this sequential fine-tuning scenario, our method also outperforms other model merging methods. We implement Task Arithmetic (Ilharco et al., 2023) (blue square) and TIES (Yadav et al., 2023) (green triangle), and show that our method achieves superior performance to both of these. In the Math-then-Law fine-tuning setting, we find that both of these methods, Task Arithmetic and TIES, fail to retain Math performance completely, whereas SFA with a low enough $p$ is

able to achieve performance on par with rehearsal. Our figure values for Pythia (2.8B) can be found in Table 1 (Math and Law), and Table 4 (Math and Code). Results for Llama 2 (7B) can be found in Table 3 (Math and Law), and Table 5 (Math and Code).

Finally, to see how our method scales as the number of domains increases, we also continue fine-tuning and applying SFA on our model for 3 domains (Fig. 4). In these graphs, we take a high performing SFA model ($p$ of 0.25) on Math and Law, and Math and Code from Fig. 3, and continue fine-tuning the model with SFA on the final domain (Code and Law respectively). We find that by using SFA (specifically adjusting $p$), we are able to maintain high performance on the previous 2 domains while also learning an additional domain. As such, SFA is a useful forgetting mitigation technique for continual learning given a sequence of domains. In both scenarios, Math-Code to Law, and Math-Law to Code, SFA (orange curve) outperforms Task Arithmetic, and sequential fine-tuning. In the case of Math-Code to Law, SFA with $p$ of 0.25 yields performance comparable to rehearsal (pink diamond). The figure results of Pythia (2.8B) fine-tuning on Math-Code to Law, and Math-Law to Code can be found in Table 6.

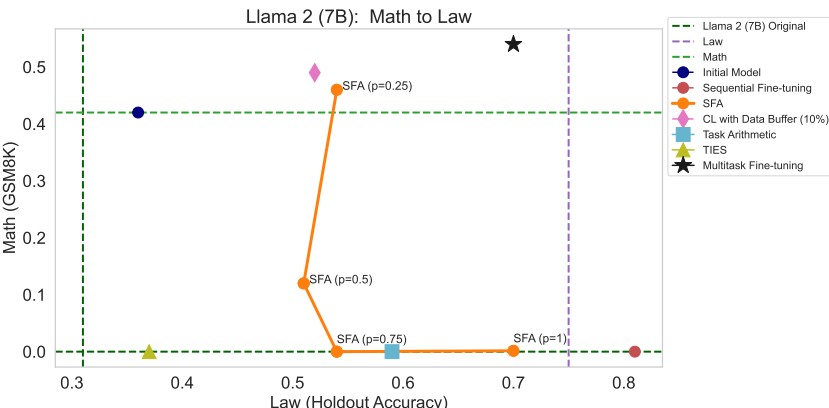

Figure 2: A comparison of Llama 2 (7B)'s performance on Math (y-axis) and Law (x-axis) using various fine-tuning and model merging techniques. The results are contained by dashed boundary boxes: the left and bottom lines represent the performance of a pretrained Llama 2 (7B) on Math and Law, whereas the right and top lines represent the performance of Llama 2 (7B) after fine-tuning on Law and Math respectively. A curve shows the performance of SFA with varying $p$, next to comparisons of continual learning with a data buffer, Task Arithmetic, and TIES. Finally, we also show an initial model (fine-tuned on math) and performance after sequentially fine-tuning it on Law.

### 6.3 L2-DISTANCE AND ACCURACY

We previously show how SFA approximates applying an L2 penalty. In order to further explore this intuition of SFA and its relation to constraining parameter weights, we also show how accuracy and L2 distance are correlated. We use the setup described in Fig. 3 where our model first fine-tunes on Math, then Law. As Fig. 5 shows, when proportion of fine-tuning before averaging $p$ decreases on SFA (purple curve), the L2 distance to the initial Math model decreases, while the accuracy on Math increases. This is in direct contrast to sequential fine-tuning without intervention (black pentagon), because of its much higher L2 distance to the initial model. As such, $p$ directly relates to L2 distance, as well as performance on previous tasks, because averaging frequency constrains how much model parameters can change from their initial positions. The values for this figure can be found in Table 2.

### 6.4 AVERAGING WEIGHTS

To further understand the advantages of SFA, we investigate alternative strategies of manipulating model parameter weights. Unlike the continuous averaging throughout fine-tuning employed by SFA, we explore the impact of modifying weights solely at the final stage. Our results underscore the importance of SFA's *continual* averaging approach for achieving optimal performance across multiple domains.

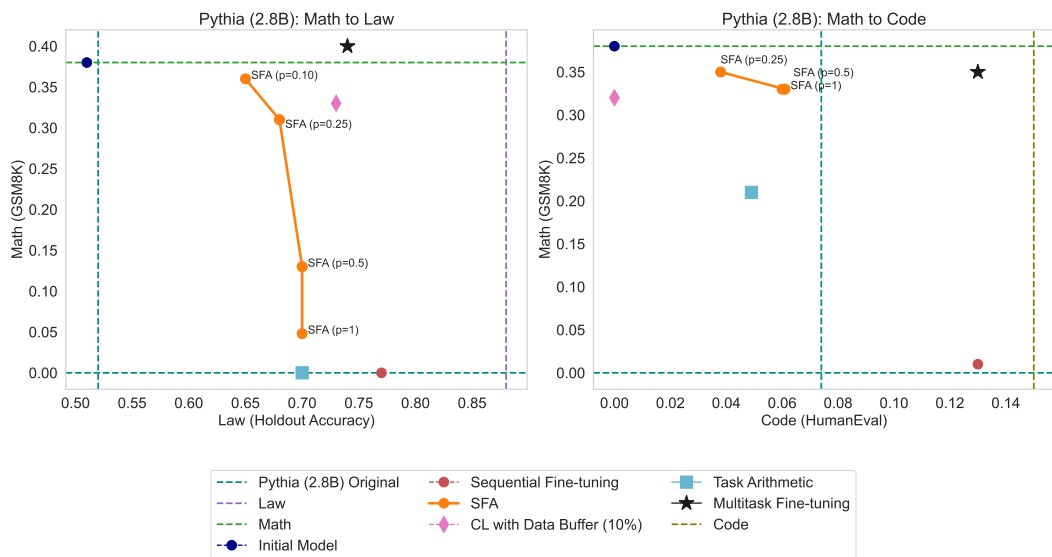

Figure 3: A comparison of Pythia (2.8B)'s performance on multiple domains (Math, Law and Math, Code) using various fine-tuning and model merging techniques similar to Fig. 2. On Math to Law, SFA $p = 0.25$ can be seen as having comparable performance to using a data buffer, while outperforming Task Arithmetic. Likewise, in Math to Code, SFA with varying $p$ outperform using a data buffer and Task Arithmetic.

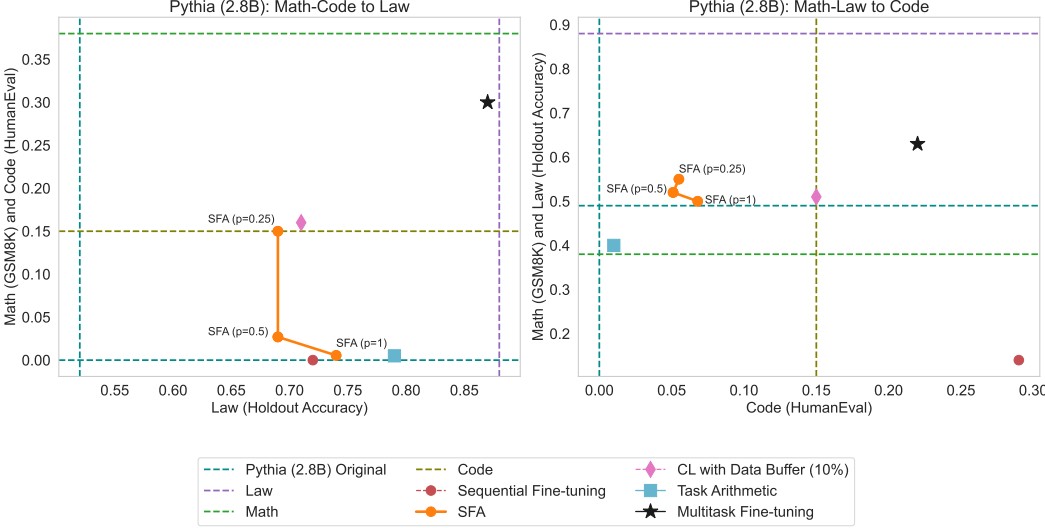

Figure 4: A comparison of Pythia (2.8B)'s performance when training on more than 2 domains (e.g. Math-Law and Code, Math-Code and Law) using various fine-tuning and model merging techniques similar to Fig. 3. On Math-Code to Law, SFA $p = 0.25$ can be seen as having comparable performance to using a data buffer, while outperforming Task Arithmetic. While, SFA with varying $p$ on Math-Law to Code outperforms Task Arithmetic, but performs worse than using a data buffer.

Recall that SFA combines parameters from the initial and current model during fine-tuning. We posit that the initial model represents expertise in past tasks/domains, while the current model embodies new task/domain knowledge. Our default parameter weighting (0.50 for each) provides a balance. We explore if, instead of varying $p$, the frequency of averaging in SFA, we can get similar flexibility by first fine-tuning the model on a new task ($p = 1$) and then averaging the final model with the

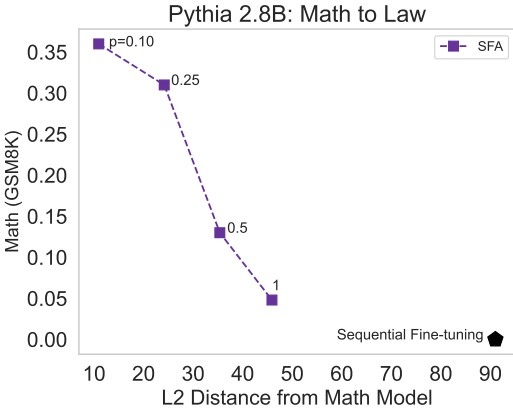

Figure 5: An analysis of the negative correlation between accuracy on Math and the $L2$ distance of the final model (fine-tuned on Math, then Law) from the original model (fine-tuned on Math only). The fine-tuning on Law is done using SFA with varying values of $p$ that determine the merging frequency. For reference we also mark sequential fine-tuning which leads to much higher L2 distance due to no merging, and accuracy just above that achieved with SFA merging once at the end of fine-tuning on law ($p = 1$).

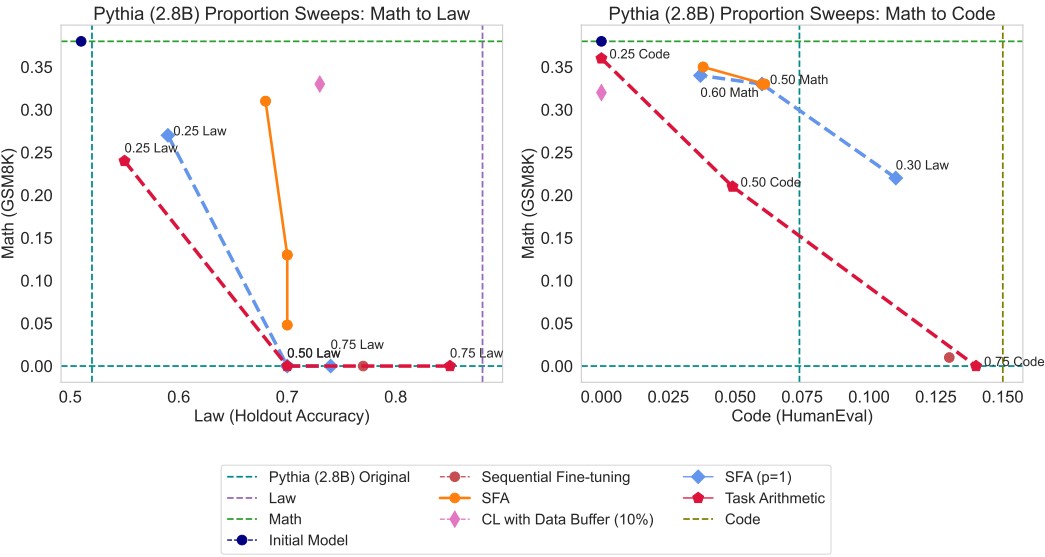

Figure 6: A comparison of varying the Task Arithmetic model weights, and $\beta$ on SFA ($p$=1), with SFA (varying $p, \beta = 0.5$) for Pythia (2.8B). We reproduce the results varying $p$ in SFA (orange curve) from Fig. 3 and add 2 sweeps showing change in performance on Pythia (2.8B) when the weights for the current and past checkpoints are varied for SFA ($p = 1$) (dashed blue) and the domain-specific models are merged in Task Arithmetic (dashed red). Generally, SFA with $p < 1$ achieves highest performance, followed by SFA ($p = 1$) with varying weights, and lastly is Task Arithmetic with varying weights.

previous task model using different relative weights (vary $\beta$). In Figs. 6 and 7, we show that SFA with $p < 1$ and $\beta = 0.5$ (orange curve) performs the same if not better than a sweep of weighting parameter $\beta$ for SFA ($p = 1$) (blue curve). Furthermore, for SFA ($p = 1$) with $\beta \geq 0.50$, the trade off between Math and Law for both Pythia (2.8B) and Llama 2 (7B) is especially large, resulting in the complete failure to retain math. This suggests that SFA's continual averaging during fine-tuning is key to its success in preserving cross-domain competence.

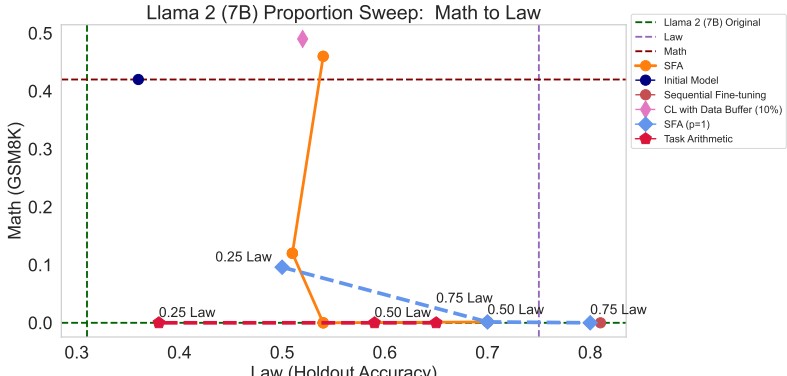

Figure 7: A comparison of varying the Task Arithmetic model weights, and $\beta$ on SFA ($p$=1), with SFA (varying $p, \beta = 0.5$) for Llama 2 (7B). We reproduce the results varying $p$ in SFA (orange curve) from Fig. 2 and add 2 sweeps for the weights on the checkpoints and domain models of SFA ($p = 1$) and Task Arithmetic, similarly to Fig. 6, to compare SFA with merging at different proportions. We see a similar outcome, where SFA with $p < 1$ generally achieves a better trade off in performance between Math and Law.

We extend this analysis to Task Arithmetic, another model merging technique. In Figs. 6 and 7 we report the results sweeping over the weight values for averaging (red curve), and observe that Task Arithmetic, like SFA ($p = 1$) with varying $\beta$, fails to achieve the cross-domain performance improvements that SFA demonstrates. Specifically, it also shows even worse combined performance on task 1 (Math, y-axis) and task 2 (Law, or Code, x-axis). Furthermore, in the Math-Law setting, for weights on Law $\geq 0.50$, it also fails to retain Math. As such, SFA $p < 1$ with $\beta = 0.50$ offers superior performance for cross domain fine-tuning on both tasks even when accounting for proportion sweeps.

## 7 CONCLUSION

In this paper, we provide a comprehensive evaluation of domain forgetting in a continual learning setting, and offer solutions to allow models to retain knowledge from all domains they fine-tune on. After showing how quickly a given model can forget learned tasks as it sequentially fine-tunes on new ones, we evaluate methods that aim to mitigate this forgetting. We introduce SFA and show how, by treating a past model as representative of past data, we can use parameter averaging to retain knowledge as the model fine-tunes on new tasks/domains. We likewise compare SFA to L2 penalty, and show how model merging methods can approximate imposing a penalty in continual learning. The final performance of SFA is comparable to continual learning with rehearsal, but has the advantage of not maintaining a data buffer. Furthermore, our solution surpasses other commonly used model merging and penalty techniques by incorporating infrequent model merging into the fine-tuning of a model.

## 8 ETHICS

This paper presents work whose goal is to advance the field of Machine Learning. There are many potential societal consequences of our work, none which we feel must be specifically highlighted here.

## 9 REPRODUCIBILITY

The tools we use in this project are all open-source. A description of our models and how we fine-tune/evaluate can be found in Appendix A.4. Descriptions of the tasks we fine-tune models on are in Appendix A.5 and Section 5. Finally, our evaluation metrics are in Table 7. We are working on releasing a repository with our specific configurations and SFA code.

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

# A  APPENDIX

## A.1  BAYESIAN INTERPRETATION

We have shown that our method approximates, and sometimes is equivalent to minimizing an L2-regression loss during training. Next we use the well known point that L2-regression has a Bayesian Interpretation (bay, 2018) to motivate our method:
Assume that the prior distribution of the ideal model $\theta_t^*$ for a past and current task is Gaussian with mean the initial model, $\theta_t^* \sim N(\theta_o, \tau^2 I)$ for some $\tau$. Furthermore, assume that the distribution $y$ given input $X$, model weights $\theta_t^*$, and a function $f$ is Gaussian with mean the output of the function given $X, \theta_t : y \sim N(f(X, \theta_t^*), \sigma^2 I)$ As such, the posterior of $\theta_t^*$ is:

$$p(\theta_t^* | y, X, f) \propto exp[\frac{-1}{2\sigma^2}(y - f(X, \theta_t^*))^T (y - f(X, \theta_t^*)) - \frac{-1}{2\tau^2}(\theta_t^* - \theta_o)^T (\theta_t^* - \theta_o)] \quad (8)$$

We can compute the Maximum a Posteriori (MAP) for $\theta_t^*$:

$$\hat{\theta}_t^* = argmax_{\theta_t^*} exp[\frac{-1}{2\sigma^2}(y - f(X, \theta_t^*))^T (y - f(X, \theta_t^*)) - \frac{-1}{2\tau^2}(\theta_t^* - \theta_o)^T (\theta_t^* - \theta_o)] \quad (9)$$

$$\hat{\theta}_t^* = argmin_{\theta_t^*}(y - f(X, \theta_t^*))^T (y - f(X, \theta_t^*)) + \frac{\sigma^2}{\tau^2}(\theta_t^* - \theta_o)^T (\theta_t^* - \theta_o) \quad (10)$$

Set $\frac{\sigma^2}{\tau^2} = \lambda$

$$\hat{\theta}_t^* = argmin_{\theta_t^*}(y - f(X, \theta_t^*))^T (y - f(X, \theta_t^*)) + \lambda(\theta_t^* - \theta_o)^T (\theta_t^* - \theta_o) \quad (11)$$

As such, L2-regression tries to solve this Bayesian interpretation (Equation 11). As shown previously, SFA approximates L2-regression. This suggests that SFA may have a Bayesian motivation.

## A.2  EWC APPROXIMATED BY MODEL MERGING

Consider fine-tuning a model with an EWC penalty (Kirkpatrick et al., 2017) where $\lambda = 1, j = 1, ..., |\theta|$

$$L(\theta_t) = L_{\text{task}}(\theta_t) + \sum_j \frac{1}{2} F_o^{(j)} (\theta_t^{(j)} - \theta_o^{(j)})^2 \quad (12)$$

where $\theta_o$ and $\theta_t$ are the weights of the initial and fine-tuning model respectively. $\eta$ is a hyperparameter, and $F_o$ is a diagonal matrix with the initial model's Fisher information. Assume that this loss update is split into 2 model updates. First, update model parameters using task loss on current weights:

$$\theta_{t+1}^* = \theta_t - \eta \Delta_{\theta_t} L_{\text{task}} \quad (13)$$

Then, update model parameters using EWC penalty:

$$\theta_{t+1} = (I - \eta F_o)\theta_{t+1}^* + \eta F_o \theta_o \quad (14)$$

Thus, applying the EWC penalty can be understood as model merging weighted by the Fisher information of the initial model. This is reminiscent of Fisher model merging from Matena & Raffel (2022) where merging an initial and fine-tuning model has the form:

$$\theta^{*(j)} = \frac{\lambda_o F_o^{(j)} \theta_o^{(j)} + \lambda_t F_t^{(j)} \theta_t^{(j)}}{\lambda_o F_o^{(j)} + \lambda_t F_t^{(j)}} \quad (15)$$

which can be rewritten as

$$\theta^{*(j)} = \left(1 - \frac{\lambda_o F_o^{(j)}}{\lambda_o F_o^{(j)} + \lambda_t F_t^{(j)}}\right)\theta_t^{(j)} + (\frac{\lambda_o F_o^{(j)}}{\lambda_o F_o^{(j)} + \lambda_t F_t^{(j)}})\theta_o^{(j)}. \quad (16)$$

Unlike the EWC approximation, this uses the Fisher information of both the initial and current model for merging.

## A.3 Forgetting under Sequential Fine-tuning

We start by confirming that fine-tuning on a sequence of different tasks leads to performance degradation on previously learned tasks. This forgetting phenomenon occurs across different task domains and for different model sizes. In this work, we focus on catastrophic forgetting of capabilities acquired during instruction fine-tuning instead of base pretrained model capabilities. This is because, as we will show, forgetting of skills learned during instruction finetuning can be quite severe and experiments at this scale are more feasible. We fine-tune our models on a sequence of instruction, language generation datasets that test general knowledge to measure forgetting. Specifically, we use Scialom et al. (2022)'s: Text Simplification (Simpl), Inquisitive Question Generation (InqQG), Headline Generation with Constraint (HGen), COVID-fact, Covid QA (CQA), and Twitter Stylometry (TwSt). Many of these tasks incorporate existing datasets which we describe in Appendix A.5. In our first experiments, we fine-tune the T0_3B (3B) and T0pp (11B) models (see Appendix A.4 for model descriptions) on the sequence of tasks described in Section 5 while measuring forgetting on the first task. The results are shown in Fig. 8. The model is first trained on Simpl which leads to a decrease in validation loss shown in blue. Subsequently, the model is trained on a sequence of other tasks; the decrease in validation loss on these tasks is shown in different colors. During this process, we continue to monitor the validation loss on Simpl, displayed in pink. As models fine-tune on new tasks, their performance on Simpl consistently declines as loss increases. This is true at both the 3B and 11B (Fig. 8) model scales, indicating that merely scaling up parameter size does not help mitigate forgetting despite the increased capacity.

But how severe is this forgetting? We quantify this by comparing a model that was trained on and has then forgotten Simpl to a model that has never seen Simpl. In Fig. 9, the pink line shows validation loss on Simpl for a model trained on a sequence of fine-tuning tasks starting with Simpl. As the model learns new tasks, its performance deteriorates. After 2000 steps, the sequentially fine-tuned model's loss on Simpl is the same order of magnitude as that of the multitask model trained on all tasks except Simpl. Thus if a model that has learned Simpl is finetuned on other tasks for as little as 2000 steps, its performance degrades to that of a model that has never seen Simpl. This indicates significant forgetting, as the model loses the ability to respond to tasks it previously was able to.

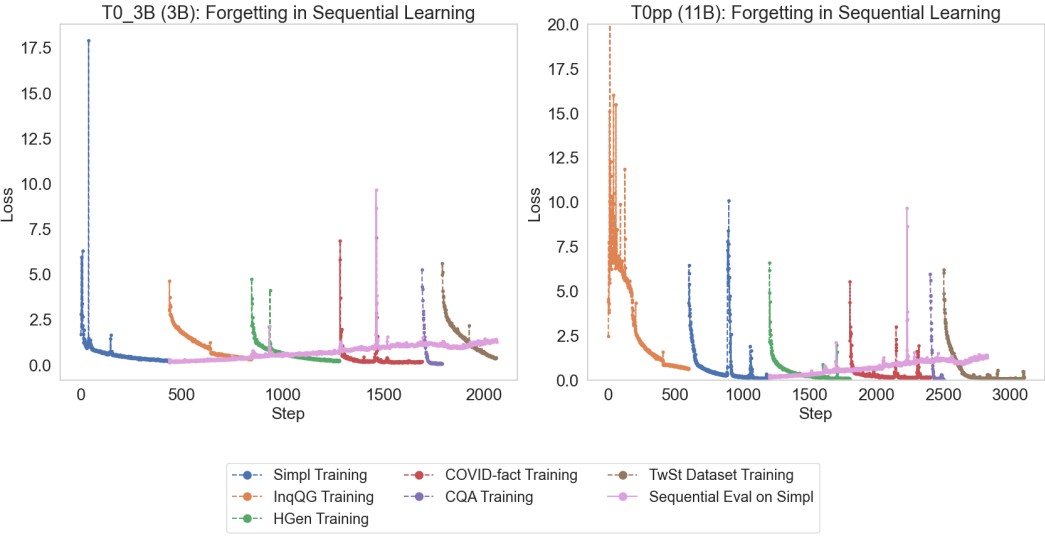

Figure 8: The fine-tuning of T0_3B (3B) and T0pp (11B) on a stream of language generation tasks. Training loss on each subsequent task decreases as the model learns it, while evaluation loss on Simpl continues to increase, indicating that forgetting is present.

To summarize, we see a consistent trend of forgetting knowledge: as models are sequentially finetuned on new tasks, performance on past tasks drops resulting in lower evaluation metrics. This gets worse as more tasks are added and is not mitigated by model scale. We will show that forgetting is even stronger when there is a domain shift between consecutive tasks (e.g. Math to Law or Code).

Table 1: Results of Pythia (2.8B) models fine-tuning on Math and Law

| PYTHIA (2.8B) | CASE_HOLD | TOS | OVERRULING | GSM8K (0-SHOT) |
|---|---|---|---|---|
| PYTHIA (2.8B) ORIGINAL | 0.25 | 0.85 | 0.45 | 0 |
| METAMATHQA | 0.19 | 0.87 | 0.48 | 0.38 |
| LAW | 0.74 | 0.93 | 0.97 | 0 |
| METAMATHQA, LAW | 0.76 | 0.95 | 0.59 | 0 |
| METAMATHQA, LAW (P=1) | 0.74 | 0.88 | 0.49 | 0.048 |
| METAMATHQA, LAW (P=1, 0.75 LAW, 0.25 MATH) | 0.78 | 0.93 | 0.52 | 0 |
| METAMATHQA, LAW (P=1, 0.25 LAW, 0.75 MATH) | 0.42 | 0.87 | 0.49 | 0.27 |
| METAMATHQA, LAW (P=0.5) | 0.69 | 0.89 | 0.52 | 0.13 |
| METAMATHQA, LAW (P=0.25) | 0.67 | 0.87 | 0.49 | 0.31 |
| METAMATHQA, LAW (P=0.10) | 0.59 | 0.87 | 0.49 | 0.36 |
| TASK ARITHMETIC (0.5 LAW, 0.5 MATH) | 0.68 | 0.87 | 0.55 | 0 |
| TASK ARITHMETIC (0.75 LAW, 0.25 MATH) | 0.73 | 0.88 | 0.95 | 0 |
| TASK ARITHMETIC (0.25 LAW, 0.75 MATH) | 0.30 | 0.87 | 0.49 | 0.24 |
| MULTITASK | 0.76 | 0.87 | 0.58 | 0.40 |
| CONTINUAL LEARNING (DATA BUFFER 10%) | 0.72 | 0.93 | 0.54 | 0.33 |

Table 2: The L2 distance of Pythia (2.8B) models from previous checkpoints of models fine-tuning on Math and law

| PYTHIA (2.8B) | L2-DISTANCE |
|---|---|
| METAMATHQA, LAW - METAMATHQA | 90.99 |
| METAMATHQA, LAW (P=1) - METAMATHQA | 45.50 |
| METAMATHQA, LAW (P=0.5) - METAMATHQA | 35.38 |
| METAMATHQA, LAW (P=0.25) - METAMATHQA | 24.16 |
| METAMATHQA, LAW (P=0.10) - METAMATHQA | 10.94 |
| TASK ARITHMETIC (0.5 LAW, 0.5 MATH)- METAMATHQA | 101.77 |
| MULTITASK - METAMATHQA | 178.96 |
| CONTINUAL LEARNING (DATA BUFFER 10%) - METAMATHQA | 82.21 |

Table 3: Results of Llama 2 (7B) models fine-tuning on Math and law

| LLAMA 7B | CASE_HOLD | TOS | OVERRULING | GSM8K (0-SHOT) |
|---|---|---|---|---|
| LLAMA 2 (7B) ORIGINAL | 0.32 | 0.13 | 0.49 | 0 |
| METAMATHQA | 0.21 | 0.38 | 0.49 | 0.42 |
| LAW | 0.81 | 0.51 | 0.94 | 0 |
| METAMATHQA, LAW | 0.64, | 0.86 | 0.93 | 0 |
| METAMATHQA, LAW (P=1) | 0.61 | 0.59 | 0.90 | 0.0015 |
| METAMATHQA, LAW (P=1, 0.75 LAW, 0.25 MATH) | 0.64 | 0.83 | 0.94 | 0 |
| METAMATHQA, LAW (P=1, 0.25 LAW, 0.75 MATH) | 0.55 | 0.16 | 0.79 | 0.096 |
| METAMATHQA, LAW (P=0.75) | 0.53 | 0.13 | 0.97 | 0 |
| METAMATHQA, LAW (P=0.5) | 0.50 | 0.13 | 0.90 | 0.12 |
| METAMATHQA, LAW (P=0.25) | 0.53 | 0.13 | 0.95 | 0.46 |
| METAMATHQA, LAW (P=0.17) | 0.48 | 0.13 | 0.63 | 0.48 |
| TASK ARITHMETIC (0.5 LAW, 0.5 MATH) | 0.68 | 0.13 | 0.96 | 0 |
| TASK ARITHMETIC (0.75 LAW, 0.25 MATH) | 0.79 | 0.18 | 0.97 | 0 |
| TASK ARITHMETIC (0.25 LAW, 0.75 MATH) | 0.44 | 0.13 | 0.56 | 0 |
| TIES | 0.37 | 0.13 | 0.61 | 0.014 |
| MULTITASK | 0.86 | 0.27 | 0.97 | 0.54 |
| CONTINUAL LEARNING (DATA BUFFER 10%) | 0.46 | 0.13 | 0.96 | 0.49 |

Table 4: Results of Pythia (2.8B) models fine-tuning on Math and code

| PYTHIA (2.8B) | HUMANEVAL (5-SHOT) | GSM8K (0-SHOT) |
|---|---|---|
| ORIGINAL PYTHIA (2.8B) | 0.074 | 0 |
| METAMATHQA | 0.0 | 0.38 |
| MAGICODER-EVOL-INSTRUCT-110K | 0.15 | 0 |
| METAMATHQA, MAGICODER-EVOL-INSTRUCT-110K | 0.13 | 0.01 |
| METAMATHQA, MAGICODER-EVOL-INSTRUCT-110K (P=1) | 0.06 | 0.33 |
| METAMATHQA, MAGICODER-EVOL-INSTRUCT-110K (P=1, 0.3 MATH, 0.7 CODE) | 0.11 | 0.22 |
| METAMATHQA, MAGICODER-EVOL-INSTRUCT-110K (P=1, 0.6 MATH, 0.4 CODE) | 0.037 | 0.34 |
| METAMATHQA, MAGICODER-EVOL-INSTRUCT-110K (P=1, 0.7 MATH, 0.3 CODE) | 0.018 | 0.38 |
| METAMATHQA, MAGICODER-EVOL-INSTRUCT-110K (P=0.5) | 0.061 | 0.33 |
| METAMATHQA, MAGICODER-EVOL-INSTRUCT-110K (P=0.25) | 0.038 | 0.35 |
| TASK ARITHMETIC (0.5 CODE, 0.5 MATH) | 0.049 | 0.21 |
| TASK ARITHMETIC (0.75 CODE, 0.25 MATH) | 0.14 | 0 |
| TASK ARITHMETIC (0.25 CODE, 0.75 MATH) | 0 | 0.36 |
| MULTITASK | 0.13 | 0.35 |
| CONTINUAL LEARNING (DATA BUFFER 10%) | 0 | 0.32 |

Table 5: Results of Llama 2 (7B) models fine-tuning on Math and code

| LLAMA 2 (7B) | HUMANEVAL (5-SHOT) | GSM8K (0-SHOT) |
| --- | --- | --- |
| LLAMA 2 (7B) ORIGINAL | 0.15 | 0 |
| METAMATHQA | 0 | 0.55 |
| MAGICODER-EVOL-INSTRUCT-110K | 0.35 | 0 |
| MAGICODER-EVOL-INSTRUCT-110K, METAMATHQA | 0.046 | 0.54 |
| MAGICODER-EVOL-INSTRUCT-110K, METAMATHQA (P=1) | 0.18 | 0.49 |
| MAGICODER-EVOL-INSTRUCT-110K, METAMATHQA (P=0.75) | 0.22 | 0.41 |
| MAGICODER-EVOL-INSTRUCT-110K, METAMATHQA (P=0.5) | 0.17 | 0.44 |
| MAGICODER-EVOL-INSTRUCT-110K, METAMATHQA (P=0.25) | 0.22 | 0.36 |
| TASK ARITHMETIC | 0.19 | 0.44 |
| TIES | 0.27 | 0.090 |
| MULTITASK | 0.09 | 0.40 |

Table 6: Results of Pythia (2.8B) models fine-tuning on Math, Law and Code for 2 orders

| PYTHIA (2.8B) | CASE_HOLD | TOS | OVERRULING |
| --- | --- | --- | --- |
| PYTHIA (2.8B) ORIGINAL | 0.25 | 0.85 | 0.45 |
| METAMATHQA | 0.19 | 0.87 | 0.48 |
| LAW | 0.74 | 0.93 | 0.97 |
| MAGICODER-EVOL-INSTRUCT-110K | 0.22 | 0.28 | 0.52 |
| METAMATHQA, LAW, MAGICODER-EVOL-INSTRUCT-110K | 0.30 | 0.87 | 0.51 |
| METAMATHQA, LAW, MAGICODER-EVOL-INSTRUCT-110K (P=1) | 0.50 | 0.88 | 0.59 |
| METAMATHQA, LAW, MAGICODER-EVOL-INSTRUCT-110K (P=0.5) | 0.55 | 0.88 | 0.57 |
| METAMATHQA, LAW, MAGICODER-EVOL-INSTRUCT-110K (P=0.25) | 0.57 | 0.88 | 0.67 |
| METAMATHQA, MAGICODER-EVOL-INSTRUCT-110K, LAW | 0.73 | 0.93 | 0.49 |
| METAMATHQA, MAGICODER-EVOL-INSTRUCT-110K, LAW (P=1) | 0.75 | 0.87 | 0.60 |
| METAMATHQA, MAGICODER-EVOL-INSTRUCT-110K, LAW (P=0.5) | 0.70 | 0.88 | 0.49 |
| METAMATHQA, MAGICODER-EVOL-INSTRUCT-110K, LAW (P=0.25) | 0.68 | 0.88 | 0.51 |
| TASK ARITHMETIC (0.33 MATH, 0.33 LAW, 0.33 CODE) | 0.63 | 0.87 | 0.88 |
| MULTITASK | 0.80 | 0.88 | 0.93 |
| CONTINUAL LEARNING (METAMATHQA, MAGICODER-EVOL-INSTRUCT-110K, LAW) (DATA BUFFER 10%) | 0.75 | 0.89 | 0.49 |
| CONTINUAL LEARNING (METAMATHQA, LAW ,MAGICODER-EVOL-INSTRUCT-110K) (DATA BUFFER 10%) | 0.69 | 0.89 | 0.56 |

| PYTHIA (2.8B) | GSM8K (0-SHOT) | HUMANEVAL (5-SHOT) |
| --- | --- | --- |
| PYTHIA (2.8B) ORIGINAL | 0 | 0.0 |
| METAMATHQA | 0.38 | 0 |
| LAW | 0 | 0 |
| MAGICODER-EVOL-INSTRUCT-110K | 0 | 0.15 |
| METAMATHQA, LAW, MAGICODER-EVOL-INSTRUCT-110K | 0.01 | 0.14 |
| METAMATHQA, LAW, MAGICODER-EVOL-INSTRUCT-110K (P=1) | 0.34 | 0.068 |
| METAMATHQA, LAW, MAGICODER-EVOL-INSTRUCT-110K (P=0.5) | 0.37 | 0.051 |
| METAMATHQA, LAW, MAGICODER-EVOL-INSTRUCT-110K (P=0.25) | 0.39 | 0.055 |
| METAMATHQA, MAGICODER-EVOL-INSTRUCT-110K, LAW | 0.0 | 0.00 |
| METAMATHQA, MAGICODER-EVOL-INSTRUCT-110K, LAW (P=1) | 0.011 | 0 |
| METAMATHQA, MAGICODER-EVOL-INSTRUCT-110K, LAW (P=0.5) | 0.054 | 0 |
| METAMATHQA, MAGICODER-EVOL-INSTRUCT-110K, LAW (P=0.25) | 0.30 | 0.0012 |
| TASK ARITHMETIC (0.33 MATH, 0.33 LAW, 0.33 CODE) | 0 | 0.01 |
| MULTITASK | 0.38 | 0.22 |
| CONTINUAL LEARNING (METAMATHQA, MAGICODER-EVOL-INSTRUCT-110K, LAW) (DATA BUFFER 10%) | 0.30 | 0.029 |
| CONTINUAL LEARNING (METAMATHQA, LAW ,MAGICODER-EVOL-INSTRUCT-110K) (DATA BUFFER 10%) | 0.30 | 0.15 |

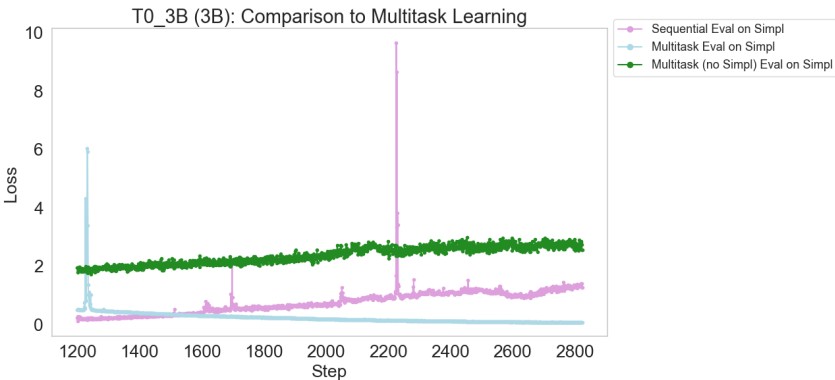

Figure 9: The Simpl loss curve of T0_3B (3B) from Fig. 8 is compared to a multitask model training on all tasks, and a multitask model training on all tasks except Simpl. As T0_3B (3B) continues to fine-tune on each new task, the loss on Simpl becomes the same order of magnitude as that of a model that is never exposed to Simpl.

Table 7: Evaluation metrics for each task and domain used in our work.

| TASK/DOMAIN | EVAL METRIC |
|---|---|
| TEXT SIMPLIFICATION (SIMPL) | TEXT SIMPLIFICATION (SIMPL) HOLDOUT SET |
| INQUISITIVE QUESTION GENERATION (INQQG) | INQUISITIVE QUESTION GENERATION (INQQG) HOLDOUT SET |
| TWITTER STYLOMETRY (TWST) | TWITTER STYLOMETRY (TWST) HOLDOUT SET |
| HEADLINE GENERATION WITH CONSTRAINT (HGEN) | HEADLINE GENERATION WITH CONSTRAINT (HGEN) HOLDOUT SET |
| COVID-FACT | COVID-FACT HOLDOUT SET |
| COVID QA (CQA) | COVID QA (CQA) HOLDOUT SET |
| LAW | CASEHOLD, TOS, OVERRULING HOLDOUT SETS |
| MATH | GSM8K (COBBE ET AL., 2021) |
| CODE | HUMANEVAL (CHEN ET AL., 2021) |

## A.4 MODELS

We fine-tune a combination of encoder-decoder and decoder only models. Specifically, we measure forgetting on T0_3B (3B) and T0pp (11B) (Sanh et al., 2021), two models already pretrained and fine-tuned on many tasks, when sequentially fine-tuning on instruction tasks (Appendix A.3). We also fine-tune Pythia (2.8B) (Biderman et al., 2023) and Llama 2 (7B) (Touvron et al., 2023) on tasks from different domains (Math, Law, Code) to measure performance on sequential learning, in addition to a variety of merging techniques (Section 6.2).

Mainly, we use Composer (Team, 2021) for fine-tuning and evaluation. For additional evaluation metrics, we also use Language Model Evaluation Harness (Gao et al., 2023). Finally, we create some model merging baselines using mergekit (Goddard et al., 2024).

## A.5 INSTRUCTION DATASETS

We use language generation tasks described in (Scialom et al., 2022) to measure forgetting. These tasks are based on pre-existing datasets that we also reference here: Text Simplification (Simpl) (Wiki-Auto (Jiang et al., 2020)), Inquisitive Question Generation (InqQG) (Eli5 (Fan et al., 2019)), Headline Generation with Constraint (HGen) (Gigaword (Graff et al., 2003; Rush et al., 2015)), Covid QA (CQA) (COVID-QA (Möller et al., 2020)), and Twitter Stylometry (TwSt) (Tweets Dataset (Bin Tareaf, 2017)).

Note: We retrieve the data for COVID-fact from (Scialom et al., 2022)'s existing codebase. We reference it using (Scialom et al., 2022) due to a lack of other citation in the paper.

