# OpenReview forum: "Soup to go: mitigating forgetting during continual learning with model averaging"
_ICLR.cc/2025/Conference — Submitted to ICLR 2025_

### Official Review · Reviewer_uFBx · 2024-10-28

**Soundness:** 2
**Presentation:** 2
**Contribution:** 2
**Rating:** 5
**Confidence:** 4

**Summary:**

This paper introduces Sequential Fine-tuning with Averaging (SFA), a method designed to mitigate catastrophic forgetting in pretrained large language models during continual learning by merging earlier checkpoints without requiring data buffers or parameter copies. SFA outperforms traditional penalty methods and merging techniques, enabling effective performance across diverse fine-tuning tasks in areas such as Math, Law, and Code.

**Strengths:**

This paper investigates an important problem on alliviating the forgetting of LLM during contiually fine-tuning. It shows that an simple technique that averages the model averages can effectively perserve the abilities of different domains.

**Weaknesses:**

While the paper presents valuable insights, I would like to point out several limitations based on my evaluation:

* **On the Novelty of Findings**: The use of model averaging (WiSE-FT [1]) to alleviate forgetting has been extensively investigated in prior work [1]. Although this paper claims to extend WiSE-FT to multiple domains, it appears to still average the model checkpoints before and after fine-tuning a new domain, which closely resembles the approach in [1, 2]. Furthermore, many of the results are consistent or overlapped with those found in [2].

* **On the Effectiveness of Model Averaging**: The paper posits that model averaging mitigates forgetting by mimicking L2 regularization towards θ0. However, numerous studies [1, 2] indicate that model averaging is generally more effective than L2 regularization. Therefore, this explanation might not sufficiently account for the observed effectiveness of model averaging.

In conclusion, neither the results nor the explanations presented seem particularly novel. Additionally, I would recommend that the paper include a more thorough discussion of existing work, particularly [2], due to the significant overlap in findings.

**References**:
[1] Mitchell Wortsman, et al., Robust Fine-Tuning of Zero-Shot Models
[2] Yong Lin et al., Mitigating the Alignment Tax of RLHF

**Questions:**

See above

---

> ### Author Response · Authors · 2024-11-25
>
> Thank you for reading the paper and providing a point on novelty that we would be eager to clarify. We want to make it clear that the novelty of our method is not in the pure extension of model merging to multiple domains. Our actual merging method is novel, because it introduces an aspect of averaging that we believe has previously not been discussed: its timing. WiSE-FT (and other common merging techniques) average 2 finished models (whether pretrained and fine-tuned, or 2 fine-tuned, etc.). Our method averages models during training and explores the role of this averaging frequency hyperparameter p. In this way, we introduce averaging as a way to change the course of an active training model’s parameter direction, and in this way connect model merging to classical continual learning methods which rely on a penalty. Furthermore, the unique case of averaging only at the end of fine-tuning (p=1) is equivalent to WiSE-FT.  However, as our results show, our method outperforms this baseline, because across all domains, frequent averaging during fine-tuning helps improve overall model performance.
> Our method performs better than the current baselines, while questioning the existing averaging methods which view it as an after-training technique. We show it can be used before this, during training, to mitigate forgetting while also allowing the model to recover any lost performance on the current task/domain through continued training. We believe this is very novel, yet at the same time intuitive as it builds on existing literature.
>
> We completely agree and show that model averaging can be more effective than L2-regression. We wanted to offer this comparison to show that model averaging can approximate L2-regression, rather than completely mimic it. However, we agree that it would be beneficial to show how they differ, and thus explain why one outperforms the other. We updated our paper to include a new version of Section 4, which better shows how SFA differs from L2-regression.

---

> > ### Comment · Reviewer_uFBx · 2024-11-27
> >
> > Thank you for the authors' response for addressing my concerns. I have raised the score accordingly.

---

### Official Review · Reviewer_evVQ · 2024-11-02

**Soundness:** 1
**Presentation:** 3
**Contribution:** 2
**Rating:** 5
**Confidence:** 5

**Summary:**

This paper concentrates on catastrophic forgetting of pretrained large language model where data from instruction fine-tuning (IFT) tasks arrives in a sequence. To mitigate the forgetting, this paper proposes Sequential Fine-tuning with Averaging (SFA), a method that merges models with earlier checkpoints trained on previous tasks during the course of training. This paper also tries to bridge L2 penalty with model merging technique, which aims to explain why the proposed method works. Finally, experiments are provided to empirically evaluate the proposed method.

**Strengths:**

1. This paper is well-written and easy to follow.

2. I appreciate the extensive experiments.

**Weaknesses:**

My main concern is about the proposed method.

1. I think the entire work (including the proposed method) is based on eq.3 which is not correct. Section 4 of this paper try to explain why the proposed method is reasonable by eq.3. According to eq.1, we can obtain the update rule of parameters, i.e. $\theta_{t+1}=\theta _t-\eta(\Delta _{\theta _t}L _{\mathrm{task}} + \theta _t- \theta _o)$. Splitting this update into 2 steps, we obtain eq.2 and $\theta _{t+1}=\theta _{t+1}^*-\eta(\theta _t- \theta _o)$. It is noteworthy that $\theta _{t+1}=\theta _{t+1}^*-\eta(\theta _t- \theta _o)$ is not equal to eq.3, because $\theta _{t+1}^*=\theta _t-\eta \Delta _{\theta _t}L _{\mathrm{task}} \neq \theta _t$. Therefore, eq.3 is not correct.

2. The issue in 1 can not be ignored, because it means that the proposed method in Algorithm 1 is not reasonable. It is a significant mistake.

**Questions:**

1. What is $p$ and $T$ in Algorithm 1? Please provide a more detailed description on the steps of algorithm.

---

> ### Author Response · Authors · 2024-11-25
>
> Thank you for showing us where our work was confusing. We would like to clarify this misunderstanding:
> The update rule we provide aims to break down the L2-regression loss into 2 gradient steps, rather than 2 steps of 1 gradient. We did this in order to mimic SFA (which typically runs a number of gradient steps in the direction of the task loss, before computing an averaging step). As such, Equations 2 and 3 correctly correspond to this by taking 1 step in the direction of the loss, and 1 step that is equivalent to merging the models. As such, we were not attempting to solve the gradient of the original L2-regression. However, we agree that our description can be easily misunderstood, and as such we replaced it with a new Section 4, as well as a Bayesian interpretation for further clarity.
>
>
> We also want to clarify that the entire work is not motivated by equation 3. This section which compares L2-regression with our method is just to provide intuition for why our method works by showing how it approximates L2-regression. Previous work on model merging and our experiments show its efficacy as a technique for mitigating forgetting, where our method naturally extends other merging literature. However, by including this section, we connect model merging to penalty based methods which are potentially more common in continual learning, in a novel way. However, we understand the confusion of this section’s placement, and will move it to the end of the paper so that this is more obvious.

---

> > ### Comment · Reviewer_evVQ · 2024-11-26
> >
> > Thanks for your detailed rebuttal. I still have some concerns.
> >
> > First of all, after checking the revision, I think the changes in Section 4 of the revision are to fix the mistakes, but not to clarify the confusion. In the initial paper, the two gradient steps of SFA are not equivalent to the gradient step of L2-regression, and cannot be regarded as an approximation of the gradient step of L2-regression too. Therefore, for the sake of rigor this is a mistake. Nevertheless, I am glad to see that my comments can help  improve this paper.
> >
> > I also need to confirm some notations and sentences. What is $T$ in Algorithm 1? In my view, $T$ is the number of gradient steps when learning a task. What is "Update model parameters θt at each time step t" in Algorithm 1? In my view, this sentence corresponds to eq.4 of the revision. These questions are stated in my initial comments, but are not answered by authors.
> >
> > I think the content of chapter 4 is not satisfactory, which aims to make the proposed method reasonable. I summarize my points as follows.
> >
> > 1. I think that SFA is not a good approximation to L2-regression. L2-regression performs step 2 in every gradient step ,i.e. $T$ times, which is usually huge in practice. SFA  performs step 2 in every $pT$ iterations, which means that the corresponding number of step 2 in SFA is $1/p$. This is a huge difference. For example, in the experiments, I notice that the minimum $p=0.1$.   This means that SFA  performs step 2 only 10 times, while $T$ is usually huge. Therefore, I think that SFA is not a good approximation to L2-regression. To be rigorous, authors need to provide approximation guarantee to analyze how close is SFA to L2-regression.
> >
> > 2. This paper concentrates on comparing SFA with L2-regression. But if I have checked correctly, L2-regression is not a compared baseline in the experiments. For the sake of rigor, this paper needs to consider L2-regression as a baseline in the experiments.
> >
> > 3. By the context of Section 4, the performance of SFA should be theoretically upper bounded by the performance  of L2-regression. However, to my knowledge, L2-regression is not well-performed compared with other classical CL methods.
> >
> > Overall, I think this paper is interesting but not rigorous. After careful consideration, I give rating 5.

---

> ### Author Response · Authors · 2024-12-01
>
> Thank you for your additional response and for helping us improve our paper, we would be eager to clarify further, and address the questions and points in order:
>
> - You mention that “In the initial paper, the two gradient steps of SFA are not equivalent to the gradient step of L2-regression, and cannot be regarded as an approximation of the gradient step of L2-regression too. “
>
> We apologize if we have not fully clarified our approach, in our opinion this is not quite what we do. In the initial paper, we take the original L2-regression loss and assume a situation where we perform 2 gradient updates instead of 1: one with the first term, and one with the second. We then show how this 2nd gradient step, which refers to imposing a penalty, can be viewed as model merging. These 2 steps are not 2 gradient steps of SFA, but rather 2 steps from the original L2-regression that have been, in this scenario, split into 2 gradient steps. We do this in order to show how purely imposing a penalty is similar to merging models. In SFA, we take some gradient steps, and then merge models. As such, we create this scenario in order to show the resemblance between SFA and a 2 step version of L2-regression. We think that strictly speaking this is not incorrect. However, we recognize that it can be misunderstood as us claiming equivalence between L2-regression and SFA, as such we updated this in the new version of the paper.
>
> - We apologize for not answering this question previously: T is the total number fine-tuning steps on a task. Updating the parameter models corresponds to Equation 4.
>
> 1) We have revised the paper to reduce the importance of SFA’s comparison with L2-regression. We did not intend for Section 4 to be one of the primary contributions of the paper. Rather, Section 4 is to provide some intuition for why model averaging might help fine-tuning performance, and not to suggest there is an exact equivalence between L2-regression and model averaging, nor that all of the good performance of SFA can be explained from its resemblance to L2-regression. Nevertheless, it may be helpful because it can provide intuition for why SFA and other model merging techniques work. Specifically, we agree that we take an extreme case of SFA, where  averaging occurs after each gradient step, and we compare this to using L2-regression, because it is the closest case of SFA to L2-regression. We show that the weight updates of these 2 cases resemble each other, and can be equivalent under certain conditions.
>
> The main justification of SFA lies in its extension of other effective merging methods (Section 3), as well as our experiments where it outperforms existing baselines (Sections 4 and 6).
>
> However, we agree that in practice, the benefit of SFA is in its infrequent averaging, and thus while this case shows the resemblance we are trying to highlight, we propose to add a more general case (e.g. averaging after a few gradient steps), to show how SFA moves away from L2-regression as averaging becomes more infrequent.
> We would be happy to add this to the final paper to add more rigor to the section.
>
> 2) We do have L2-regression as a baseline. See Figure 1.
>
> 3) As we show in Figure 1, SFA outperforms the baseline of L2-regression. This is consistent with CL literature, and with the contents of Section 4. It was not our intention to suggest that L2 is the gold standard and that any good performance of SFA is due to its approximation of L2-regression. Instead, our intention is to suggest that some intuition behind the good performance of SFA might be explained by its resemblance to L2-regression. However, the fact that SFA outperforms L2-regression, does indeed suggest that there is probably something else at work that might explain performance differences. That is, the way in which SFA differs from L2-regression seems to work  in favor of SFA. Nevertheless, we think the comparison is helpful, much in the way that one might explain the performance of L2-regression by first discussing Least Squares.

---

### Official Review · Reviewer_KE43 · 2024-11-02

**Soundness:** 2
**Presentation:** 1
**Contribution:** 2
**Rating:** 3
**Confidence:** 4

**Summary:**

This paper introduces the SFA method, designed to address the issue of catastrophic forgetting in the continuous learning process of models. The method does not rely on historical data but instead continuously merging checkpoints from the training process with the original model. SFA is equivalent to L2 regularization to some extent. In the settings proposed in the paper, this method outperforms existing penalty update methods. The paper further analyzes the performance of the SFA method under different merging frequencies.

**Strengths:**

- The method is simple and easy to understand, which could facilitate practical usage if proven effective.
- The paper compares the method with several existing approaches, validating its effectiveness.

**Weaknesses:**

- The overall writing and organization of the paper are unclear in many places. For example:
  1. The reviewer believes that the formula in Algorithm 1 should be $\theta_t = \beta \theta_0 + (1-\beta) \theta_{t-pT}$. The key point of the algorithm, as understood by the reviewer, is to mix the merged and updated model with the original model to maintain performance.
  2. The explanation from lines 154 to 156 is confusing.
  3. The experimental results are introduced starting on page 5, but the related figures are placed on page 7, making the paper difficult to read.
  4. In Figure 1, the legends for L2 and EWC should be dashed lines.
  5. In the series of figures starting from Figure 2, the author attempts to include too much comparative information in a single figure, making them hard to understand.
- The models used in the paper are all the relatively poorly performing Pythia 2.8B, leading to a direct drop in GSM8K accuracy to 0% in some cases. This limits the reliability of the results in these scenarios. The authors are encouraged to conduct experiments on more recent and higher-performing small language models.

**Questions:**

- Besides linking SFA with L2 penalty, could the reviewer further discuss why SFA significantly outperforms L2 penalty? Are there specific scenarios where one method is more suitable than the other?

---

> ### Author Response · Authors · 2024-11-25
>
> Thank you for pointing out unclear portions of the paper, we will fix all of these issues.
>
> We actually include Llama 2 7B, and a small custom image classification model, in addition to Pythia 2.8B in the experiments, as well as Appendix of this paper to offer differences in scaling and performance. For the final paper, we will also include a more updated model: Gemma 2B, and ViT (see the rebuttal for reviewer kPso).
> We believe that the drop in performance on GSM8K is important to show how well our method works when catastrophic forgetting occurs. By showing such extremes of forgetting, we only strengthen the effectiveness of our method, because if GSM8K drops to 0%, and our method improves performance so significantly, it must be effective.
>
> We have also added a more direct mathematical comparison of our method compared to L2 penalty, as well as Bayesian interpretation, to offer intuition as to why our method outperforms it in the updated version of the paper (Section 4).

---

> > ### Comment · Reviewer_KE43 · 2024-11-28
> >
> > Thanks for the authors' response. My concerns about the the experiments have been largely addressed.
> >
> > I believe there is still significant room for improvement in the layout of the experiments in the appendix and the figures in the main text. As Reviewer evVQ mentioned, the additional explanations provided by the authors are not entirely rigorous and lack appropriate baselines for comparison.
> >
> > Based on these considerations, I will maintain my current score.

---

> > > ### Author Response · Authors · 2024-12-02
> > >
> > > Thank you for your response and we are glad to have addressed your concerns about the experiments.
> > >
> > > We have addressed Reviewer evVQ’s explanation concerns in our latest response, and actually do include the appropriate baselines in Figure 1.
> > >
> > > In terms of the layout of the experiments and figures, could you please provide us with actionable suggestions as we would be happy to address any formatting issues. Thank you.

---

### Official Review · Reviewer_kPso · 2024-11-02

**Soundness:** 3
**Presentation:** 3
**Contribution:** 2
**Rating:** 5
**Confidence:** 4

**Summary:**

This paper proposes Sequential Fine-tuning with Averaging for continual learning in LLMs pre-training. The proposed method mitigates forgetting by periodically merging the fine-tuning model with earlier checkpoints trained on previous tasks.

**Strengths:**

1. The motivation for the proposed methodology is clear: the proposed method reduces forgetting without intensive memory costs for storing past data or multiple copies of previous checkpoints.
2. This paper presents empirical comparisons of the proposed method with regularization-based CL and model merging in LLMs pre-training CL. These results verify the effectiveness of the proposed method.

**Weaknesses:**

1. The proposed is validated in limited settings: it only tests on 3 or 4 tasks. Moreover, the order of arrival of these 3 or 4 tasks is fixed. This is not practical in continuous learning, where there may be an infinite number of tasks and we have no control over the incoming tasks.
2. The proposed method can be applied to CV tasks. Thus, it can be tested in CV tasks under more settings and compared with more existing works. However, this paper fails to do so.
3. The performance of the proposed method is inferior to the data replay-based method in most cases.
4. [Minor] In "Model Merging" of related works, the paragraphs have no space between them, making it uncomfortable to read.

**Questions:**

Can the proposed method be generalized to longer task sequences? If so, I would suggest that the authors use the CIFAR100 or ImageNet datasets to construct task sequences, as is done in existing CL works, and compare the proposed method to more CL baselines.

---

> ### Author Response · Authors · 2024-11-25
>
> Thank you for the comments and insights on our experimental setup. In the current setup, we actually fine-tune on 3 different domains (where each domain can include multiple tasks), on computationally large models and datasets in order to test the robustness of our method. We wanted to highlight these experiments because we show that our method works even in extreme cases as each domain tests very different knowledge and skills. Also, we want to note that fine-tuning LLMs on large language datasets, can be very computationally expensive, and that consequently it is conventional to use a smaller set of tasks than in other domains (e.g. Ilharco et al. 2023 [1] finetune on 4 tasks from the GLUE benchmark).
>
> However, we are also currently running and expect to provide more continual learning settings where a single model is trained on a stream of ~20 image classification tasks to show that our method can also be applied to CV with a larger number of tasks. We are currently running experiments using streams of tasks from Food101, and plan to also include CIFAR100 to show that our method works well on continual learning settings in CV.
>
> Our experiments tend to show that SFA performs comparably to using data replay, but without the need to store an extra data structure. This in itself is useful given the scenarios we describe where storing a data buffer to use for replay can be expensive to store, particularly because our domains include large datasets, and models.  Furthermore, selecting data to store in the data buffer from past tasks, as well as training with a combined dataset of current and past task data can also be an incredibly difficult task.
> We will fix the spacing issue in the Related Works.
>
> [1] Gabriel Ilharco, Marco Tulio Ribeiro, Mitchell Wortsman, Suchin Gururangan, Ludwig Schmidt, Hannaneh Hajishirzi, and Ali Farhadi. Editing models with task arithmetic, 2023.

---

> ### Comment · Reviewer_kPso · 2024-11-27
> **Official Comment from Reviewer kPso**
>
> Thanks for the clarification. I maintain my score.

---

### Meta-Review · Area_Chair_T7Ro · 2024-12-16

**Metareview:**

(a) summary

This paper investigates how to mitigate catastrophic forgetting during sequential finetuning of a pretrained large language model. It proposes to average the model weights with previous checkpoints for maintaining the performance on the earlier tasks. Experiments in continual learning(CL) setting demonstrates the proposed method performs better than some existing CL approaches.

(b) strengths
+ The paper is well-motivated: the proposed method reduces forgetting without intensive memory costs for storing past data or multiple copies of previous checkpoints.
+ The proposed method is both simple & effective: the empirical comparisons of the proposed method with regularization-based CL and model merging in LLMs pre-training CL verify the effectiveness of the proposed method.

(c) weaknesses
- It has limited novelty: model merging has been used for handling catastrophic forgetting in previous work:
  [1] Mitchell Wortsman, et al., Robust Fine-Tuning of Zero-Shot Models
  [2] Yong Lin et al., Mitigating the Alignment Tax of RLHF

- It has limited experimental settings: it tests only on 3 or 4 tasks, and the order of arrival of these 3 or 4 tasks is fixed. This is not practical in continuous learning, where there may be an infinite number of tasks and we have no control over the incoming tasks.
- There is not enough analysis why the proposed method is effective.
- The performance of the proposed method is inferior to the data replay-based method in most cases.
- It needs thorough discussion on related work on model merging due to significant overlap.

(d) decision

This paper addresses an important problem in CL and proposes a simple and effective method to handle forgetting in LLM finetuning. However, as pointed out by the reviewers, there is limited novelty due to the significant overlap with previous work on model averaging.  For this reason, I recommend reject.

**Additional Comments On Reviewer Discussion:**

While the reviewers acknowledged the good motivation, the simple and effective method, there are shared concerns on the novelty of the proposed method, the limited experimental settings, and inferior performance comparing with data replay-based method in most cases. The authors rebuttal and updated manuscript addressed some concerns, however, the reviewers were not convinced after the rebuttal.

---

### Decision · Program_Chairs · 2025-01-22

Reject